# Proteomic Expression Profile in Human Temporomandibular Joint Dysfunction

**DOI:** 10.3390/diagnostics11040601

**Published:** 2021-03-28

**Authors:** Andrea Duarte Doetzer, Roberto Hirochi Herai, Marília Afonso Rabelo Buzalaf, Paula Cristina Trevilatto

**Affiliations:** 1Graduate Program in Health Sciences, School of Medicine, Pontifícia Universidade Católica do Paraná (PUCPR), Curitiba 80215-901, Brazil; rherai@gmail.com (R.H.H.); paula.trevilatto@pucpr.br (P.C.T.); 2Department of Biological Sciences, Bauru School of Dentistry, University of São Paulo, Bauru 17012-901, Brazil; mbuzalaf@fob.usp.br

**Keywords:** temporomandibular joint, protein expression, temporomandibular joint dysfunction

## Abstract

Temporomandibular joint dysfunction (TMD) is a multifactorial condition that impairs human’s health and quality of life. Its etiology is still a challenge due to its complex development and the great number of different conditions it comprises. One of the most common forms of TMD is anterior disc displacement without reduction (DDWoR) and other TMDs with distinct origins are condylar hyperplasia (CH) and mandibular dislocation (MD). Thus, the aim of this study is to identify the protein expression profile of synovial fluid and the temporomandibular joint disc of patients diagnosed with DDWoR, CH and MD. Synovial fluid and a fraction of the temporomandibular joint disc were collected from nine patients diagnosed with DDWoR (*n* = 3), CH (*n* = 4) and MD (*n* = 2). Samples were subjected to label-free nLC-MS/MS for proteomic data extraction, and then bioinformatics analysis were conducted for protein identification and functional annotation. The three TMD conditions showed different protein expression profiles, and novel proteins were identified in both synovial fluid and disc sample. TMD is a complex condition and the identification of the proteins expressed in the three different types of TMD may contribute to a better comprehension of how each pathology develops and evolutes, benefitting the patient with a focus–target treatment.

## 1. Introduction

Temporomandibular dysfunction (TMD) is a disorder of the masticatory system and it is characterized by pain, loss of function of one or both articulations, and impairment of the masticatory system. TMD impacts not only jaw function, but the life quality of affected patients, increasing their treatment costs and work absence [1]. According to the National Institute of Health [2], TMD management in the USA costs approximately 4 billion dollars per year. A diagnostic protocol developed for research named Research Diagnostic Criteria for TMD (RDC/TMD), classifies TMD as myalgia, arthralgia, condylar pathologies, disc displacement, osteoarthrosis, osteoarthritis, degenerative joint disease and subluxation [3]. TMD has a multifactorial etiology, the most common being trauma, psychological alterations, hormone, inflammatory diseases, parafunction, and genetics [1,4]. TMD usually requires a panorex, and depending on the TMD type, magnetic resonance imaging, scintigraphy and tomography, besides a thorough clinical evaluation [5,6].

Depending on the TMD type, it can be classified as condylar hyperplasia (CH), disc displacement without reduction (DDWoR) and mandibular dislocation (MD). DDWoR is the most common TMD disorder [7], and along with CH, its etiology’s understanding is still unclear. MD is a condition that is probably caused by physical alterations [8], and since it is less likely to have hormone contribution, it is a good TMD condition to compare the results with the other pathologies. DDWoR is caused by an abnormal positional association between the disc and the condyle, where the disc is permanently anteriorly displaced in relation to the condyle, causing limited range of mouth opening, pain and may lead to temporomandibular joint (TMJ) degeneration [9]. Disc displacement corresponds to 41% of TMD intra-articular disorders [7], and it is considered a multifactorial disease, with overlapping conditions contributing to its modulation including stress, parafunction, behavioral pattern, emotional status, and genetic background [3]. Among its different types of treatment, clinical handling is firstly employed (splint therapy, medication, physiotherapy) and when unsuccessful, surgery is indicated [6,10]. MD is an involuntary forward movement of the condyle beyond the articular eminence, mostly associated with trauma or excessive mouth opening, impairing its essential functions (speaking, chewing), and it accounts for 3% of all documented dislocations [11]. It usually needs mechanical manipulation to return to its normal position, and recurrent dislocations require surgical treatment [8]. Between these TMD types, CH is the rarest pathology that manifests a head condyle overgrowth, causing facial asymmetry, deformity, malocclusion and sometimes pain and dysfunction [12]. It is a self-limiting condition, more prevalent in female teenagers, but it usually requires surgical treatment to limit facial asymmetry progression and condyle continuous elongation [13]. Studies suggest it has a genetic involvement on its development, but its main etiology is still poorly understood [14]. 

Despite the etiological differences between CH, DDWoR and MD, current studies have limited understanding of the molecular variations that differentiates these TMD diseases. Condylar hyperplasia, mandibular dislocation and disc displacement have been the aim of many studies, due to their difficulty in targeting the proper treatment to each disease [9]. The employment of specific treatment, which may be improved with the unveiling of its specific etiology factors, will allow us to diminish treatment time and costs. 

At the proteomic level, current studies focus only on individual mandibular dysfunctions, without comparing different TMD types to show the proteomic variability that could drive novel biomarkers as targets for disease diagnostic and treatment [15,16]. Proteomic analysis is a gold standard approach to analyze all identifiable proteins in a certain tissue, investigating its abundance, variety of proteoforms, and their stable or transient protein–protein interactions. This approach is especially beneficial in the clinical setting when studying proteins involved in different pathologies [17]. To date, there are very few studies investigating human TMD samples through proteomic output, and these studies analyzed only synovial fluid, focusing on specific target proteins [15,16]. Therefore, analyzing all proteins present in the synovial fluid and disc sample of different types of TMD may potentially lead TMD treatments towards a new reality. 

In this research, a high throughput proteomic investigation of the three TMD pathologies CH, DDWoR and MD, was performed. Using state-of-the-art sample extraction procedures, biological samples of synovial fluid and TMJ discs were collected from distinct patients diagnosed with these conditions. The samples were processed, subjected to protein extraction and mass spectrometry proteomic identification. Generated proteomic data were analyzed using bioinformatics methods, and a per-sample protein identification and annotation were performed. The clinical phenotypes were then used to correlate the proteomic profile of each TMD condition. 

## 2. Materials and Methods

### 2.1. Sample Selection

The sample was composed of 9 disc and synovial fluid specimens from female patients, with a mean age of 31.22 years (18–52). The patients presented different TMJ conditions, with three samples being composed of TMJ displaced disc without reduction (*n* = 3), two mandibular dislocation (*n* = 2) and four patients with condylar hyperplasia (*n* = 4) (Table 1). The specimens were collected from patients treated at the Evangelic University Hospital of Curitiba, Brazil. The study was approved by the Ethical Committee on Research at Pontifical Catholic University of Paraná, Brazil, according to Resolution 196/96 of the National Health Council and approved on 6 May of 2016 under registration number 1.863.521. 

Subjects did not present any of the following criteria: use of orthodontic appliances; chronic usage of anti-inflammatory drugs; history of diabetes, hepatitis, HIV infection; immunosuppressive chemotherapy; history of any disease known to compromise immune function; pregnancy or lactation; major jaw trauma; previous TMJ surgery; and previous steroid injection in the TMJ. 

Subjects answered a personal medical history questionnaire and signed a consent form after being advised of the nature of the study. All patients were clinically examined by one experienced oral and maxillofacial surgeon. The clinical examination consisted of palpating the TMJ region, analyzing the occurrence of painful or limitation/excessiveness of mouth opening/closing, and the observation of facial asymmetry. Regarding complementary exams, all patients had a panorex and patients with disc displacement were submitted to a magnetic resonance image. The patients who were considered to be affected with disc displacement were treated surgically when they presented painful clinical signs of disc displacement after unsuccessful non-surgical treatment for at least 6 months [18]. Patients presenting pain related only to muscular spasms were not included in this research. Patients with condylar hyperplasia were diagnosed through clinical evaluation, panorex and when presenting a positive condylar growth in scintilography, a high condylectomy was indicated and performed [19]. Patients with recidivist mandibular dislocation (more than four episodes in six months) were treated with eminectomy [8].

### 2.2. Sample Acquisition

During access to the TMJ to perform the needed surgery [20], a 21-gauge needle was inserted into the upper TMJ space, then 1 mL of saline was injected into the joint space, which was aspirated thereafter by a second adapted syringe. This procedure was repeated five times to obtain a synovial fluid sample as described previously by Alstergren [21]. For each type of surgery performed, TMJ disc recontouring and repositioning was needed [16], therefore, first the displaced disc was freed, repositioned and sutured to the latero-posterior side of the condyle with a Mitek bone-cleat. The suture was then placed between the posterior and intermediate bands, and recontouring the thickened disk with a scalpel was necessary (this posterior debrided cartilage constituted the disc sample). Synovial fluid was spun down at 300× *g* to remove debris, and stored at −80°C until use or analysis, and the disc samples rinsed in phosphate-buffered saline (PBS), and either snap frozen in liquid nitrogen and stored at −80°C. 

### 2.3. Proteomic Analysis

The microcentrifuge tubes containing the synovial fluid and TMJ discs were removed from the −80 ° C freezer, and after defrosting, the discs were cut into small pieces with the aid of sterile scissors, centrifuged, and the supernatants were collected and pooled according to each pathology group. The preparation of the samples for proteomic analysis was carried out as previously reported [22]. The analysis of the tryptic peptides was performed in the nanoACQUITY UPLC system (Waters, Milliford, CT, USA) coupled to the Xevo Q-TOF G2 mass spectrometer (MS) (Waters, Milliford, CT, USA). For this purpose, the UPLC nanoACQUITY system was equipped with a column of type HSS T3 (Acquity UPLC HSS T3 column 75 mm × 150 mm; 1.8 µm, Waters), previously balanced with 7% of the mobile phase B (100% ACN + 0.1% formic acid). The peptides were separated through a linear gradient of 7%–85% of the mobile phase B over 70 min with a flow of 0.35 µL/min and the column temperature maintained at 45 °C. The MS was operated in positive ion mode, with a 75 min data acquisition time. The obtained data were processed using ProteinLynx GlobalServer (PLGS) version 3.03 (Waters, Milliford, CT, USA). Protein identification was obtained using the ion counting algorithm incorporated into the software. The collected data were searched in the database of the species *Homo sapiens* downloaded from the catalog of the UniProt [23] in September of 2020. The identified proteins for the groups DDWoR, MD, and CH of synovial fluid and TMJ disc were classified and attributed by biological function, origin, and molecular interaction with the program Genemania [24]. The overlapping proteins between the groups were clustered by using an automatic Venn diagram generator.

## 3. Results

In this qualitative study, our aim was to explore, for the first time, a comparative analysis of the proteomic profile of three distinct TMD diseases. Although a statistical analysis was not performed, we were able to identify and describe the function of the proteins, including overlapping proteins between the investigated samples (DDWoR, MD and CH, and between both synovial fluid and disc samples). 

In the synovial fluid samples, a total of 225 proteins (351 counting the repeated proteins in all groups) were successfully identified: 190 in the group DDWoR, 154 in the group MD and seven in the group CH. We also compared these three groups to identify shared or condition-specific proteins. We found 114 shared proteins between groups DDWoR and MD, and six proteins were shared by all groups (Table 2). 

In the disc sample, 379 proteins were identified (697 counting the repeated proteins in all groups), with 235 proteins in group DDWoR, 196 in group MD and 266 in group CH. These three groups were also compared to identify shared or condition-specific proteins. There were nine shared proteins between groups DDWoR and MD, 28 shared proteins between groups DDWoR and CH, 17 shared proteins between groups MD and CH, and 132 shared proteins by all groups (Table 3).

Regarding the proteins in common in both synovial fluid and disc in the same sample groups, DDWoR presented two common proteins, MD presented three proteins, group CH had no protein in common, and the three groups together had six proteins in common (Table 4). 

All synovial fluid and disc samples presented proteins involved in DNA repair, muscle and neural regeneration. 

A selective pool of proteins was chosen to be studied according to the pathology group and protein function for synovial fluid and disc sample (Table 5 and Table 6). 

The synovial fluid sample presented the following proteins functions for each group (Table 5): the DDWoR group presented proteins involved in inflammatory process, apoptosis, hearing, interleukine-6 cascade, and protection against oxidative stress; the MD group showed proteins involved in inflammatory process, apoptosis, hearing, interleukine-6 cascade, protection against oxidative stress, and immune response; in the CH group, the expression of alcohol degradation protein (ADH1) was identified. The group comprising the pathologies DDWoR and MD were mainly involved in inflammatory process inhibition, bone resorption, chondrogenesis, bone and cartilage formation, osteoarthrosis, and neuropathic pain. No proteins were observed in the groups DDWoR and CH, and MD and CH. The proteins expressed in all three groups (DDWoR, MD and CH) were mainly implicated with muscle regeneration. 

The disc sample presented the following protein functions for each group (Table 6): the DDWoR group expressed proteins involved in inflammatory process, neurogenesis, cartilage formation, extracellular matrix degradation, oxidative stress and apoptosis. The MD group presented proteins related to apoptosis, vascular growth, inflammatory inhibitors, immunologic factors and epithelial growth, and the CH group showed protein expression implicated in apoptosis, apoptosis inhibition, oxidative stress, bone formation, chondroitin, bone and cartilage formation. The group with DDWoR and MD samples had proteins involved in inflammatory process; the group with DDWoR and CH samples showed proteins with collagen formation and wound healing functions; the group with MD and CH was involved in wound healing; and the group containing DDWoR, MD and CH samples was involved with inflammatory cascade modulation, osteoclastogenesis, chondrogenesis, apoptosis, bone formation, vascular and tissue repair, antioxidative activity.

There were proteins identified in both synovial fluid and TMJ disc samples, however, some of them in different pathology groups (Table 7).

Different types of collagen were identified in discs of the MD group, CH group, DDWoR and CH group, and in the group with all pathologies together (DDWoR, MD and CH). Besides the known collagen type I present in TMJ discs, collagen type IV, VI, XII and XIV were also identified (Table 8). 

All shared and group-specific proteins are indicated in a Venn diagram for the synovial fluid (Figure 1) and disc samples (Figure 2). 

The interactions between the proteins were analyzed with Genemania (https://genemania.org—accessed on 5 September 2020), and its genetic network pointed out distinct protein cascades that might be modulating each pathology through the synovial fluid and disc samples. The physical and genetic interactions, co-expression and pathway of the proteins are shown in Figure 3 and Figure 4.

The main proteins with important functions and networks that were identified in the synovial fluid sample were analyzed for each group (Figure 3). A brief description of these findings are: in the DDWoR group (Figure 3A) alpha-2-macroglobulin (A2M) involved in inflammatory process, amyloid P component (APCS) involved with apoptosis and complement factor H (CFH) that modulates inflammatory cascade were highlighted in the Genemania interaction figure; in the MD group (Figure 3B), hemopexin (HPX) involved in protection against oxidative stress was present; in the CH group (Figure 3C), alcohol dehydrogenase subunit alpha (ADH1) that is responsible for alcohol degradation and interacts with growth hormone receptor (GHR) was present. In the group of DDWoR and MD (Figure 3D), annexin A1 (ANXA1), decorin (DCN), and immunoglobulin heavy constant gamma 1 (IGHG1) involved in inflammatory process, annexin A2 (ANXA2) involved with bone resorption, asporin (ASPN), biglycan (BGN), cartilage intermediate layer protein (CILP), osteoglycin (OGN), transforming growth factor beta induced (TGFBI) involved in bone and cartilage formation, fibronectin 1 (FN1), lumican (LUM) and tenascin XB (TNXB) involved in tissue repair, and neurofilament medium (NEFM) and thrombospondin 4 (THBS4) involved in neuropathic pain were included in the net. The DDWoR and CH group, and MD and CH group had no protein to be analyzed. The group with the three pathologies (DDWoR, MD and CH) showed an interaction of enolase 2 (ENO2) and 3 (ENO3), involved in muscle regeneration (Figure 3E). 

The disc sample presented the following protein interactions in Genemania (Figure 4): group DDWoR (Figure 4A) presented mainly the matrix metalloproteinase protein (MMP) family (1,2,3,6,8,10,13,15,16), integrin subunit alpha 6 (ITGA6) and phospholipase A2 group VII (PLA2G7) that are involved in inflammatory cascade. Additionally, thrombospondin 3 (THBS3) and 4 (THBS4) involved in tissue remodeling, and THADA armadillo repeat containing (THADA) involved in apoptosis were present. In the MD group (Figure 4B), A-kinase anchor protein 13 (AKAP13), Erbin (ERBIN) and uroplakin-3a (UPK3A) involved in apoptosis, collagen alpha-1(IV) chain (COL4A1) and GTPase Eras (ERAS) involved in disc matrix constitution, and liprin-alpha-1 (PPFIA1) and (PPFIA2) 2 responsible for cell interactions were identified in the Genemania network. In the CH group (Figure 4C), the present proteins were ADAM metallopeptidase domain 10 (ADAM10), that regulates apoptosis, collagen type I alpha 2 chain (COL1A2) and serpin family H member 1 (SERPINH1) involved in collagen formation, actinin alpha 4 (ACTN4), PDZ Additionally, LIM domain 4 (PDLIM4), transthyretin (TTR) and protein tyrosine phosphatase non-receptor type 13 (PTPN13) involved in apoptosis, hormone modulation and bone formation. In the group of DDWoR and MD (Figure 4D), the complement C4A (C4A) and complement C4B (C4B) proteins that mediates the inflammatory process were identified. In the DDWoR and CH group (Figure 4E), mainly the proteins aggrecan (ACAN), collagen type I alpha 1 chain (COL1A1) and collagen type IV alpha 6 chain (COL4A6) that constitutes disc matrix, and periostin (POSTN) involved in wound healing were identified. In the MD and CH group (Figure 4F), keratin 6A (KRT6A) involved in wound healing was identified. Additionally, in the group with all three pathologies (DDWoR, MD and CH) the proteins that interacted were annexin A1 (ANXA1), complement C3 (C3) and tenascin C (TNC) involved in inflammatory cascade modulation, annexin A2 (ANXA2) and transforming growth factor beta induced (TGFBI) involved in osteoclastogenesis, asporin (ASPN), biglycan (BGN), collagen type VI alpha 1 chain (COL6A1), osteoglycin (OGN) and vimentin (VIM) involved in chondrogenesis and osteogenesis, amyloid P component (APCS) and complement C3 (C3) in apoptosis and lumican (LUM) involved in tissue repair (Figure 4G). 

## 4. Discussion

The different types of TMD may jeopardize patients’ quality of life, masticatory function and have a great impact on health expenses. The identification of its multifactorial etiological components will enhance the employment of specific treatments, diminishing the hazard it causes in the TMJ. Therefore, the identification of the proteins expressed on each pathology group of this study (DDWoR, MD, and CH) might elucidate the cascades involved in the progression and severity of each TMD, leading to an assertive handling of TMD.

A total of 225 proteins were identified in the synovial fluid sample, and 379 in the TMJ disc sample (Table 2). It is important to highlight that the synovial fluid sample is very complex to obtain, therefore some proteins might not have been identified due to the technique that advocates the dilution of the synovial fluid. Nevertheless, the sample was collected according to worldwide employed standard methods previously described by other research groups [21,25]. Additionally, even though few proteins’ expression might not have been observed, the expression of new proteins were identified for each pathology group, which enriches the global analysis of this study. 

In our analysis, we found that all proteins expressed in the DDWoR group (synovial fluid and disc sample) (Table 2 and Table 3) presented many proteins related to inflammatory process (MMP-3, -10, -27 in the disc sample) and apoptosis (mitogen-activated protein kinase 7—MAP3K7) and THADA in synovial fluid). Only the MMP-3 protein was previously associated with TMD [26,27]. These are proteins that highly impact the degeneration process in the TMJ of patients with DDWoR [26,28]. In the MD group, ERBIN protein was found in the disc sample, and it modulates TGFB, which was previously associated with TMJ degeneration [29]. Additionally, unprecedented proteins were seen in the synovial fluid associated with apoptosis (aldehyde dehydrogenase 1 family member L—ALDH1L1) and protection against oxidative stress (HPX), which probably helps diminish the mechanical overload consequences of the dislocation in the TMJ. Regarding CH proteins in the synovial fluid sample, ADH1 catalyzes the oxidation of alcohols to aldehydes, but as seen in Genemania (Figure 3C), it interacts with GHR, which might be involved with the condylar overgrowth. In a previous study, GHR has been injected in rabbits’ TMJ to increase cartilage thickness [30], but it has not been studied as a possible etiology of condylar overgrowth yet.

Additionally, we also found a set of proteins to be common in both synovial fluid and disc samples (Table 4) in the groups DDWoR (chromodomain-helicase-DNA-binding protein 8 and myosin light chain 6B), MD (filamin A and liprin-alpha-1), and in the three groups (enolase 1, 2, 3, myosin heavy chain 16, ribosomal protein L7 like 1 and component of the shield in complex). These proteins were involved in cell matrix adhesion, cellular motor protein, reorganization of cytoskeleton, muscle development and regeneration. Additionally, another group of proteins were identified in both synovial fluid and disc samples (Table 7), being prevalent in all groups of disc samples. In the DDWoR and MD groups of synovial fluid samples, proteins implicated in apoptosis, inflammatory process, bone formation and resorption, chondrogenesis, wound healing, tissue repair and protection against oxidative stress were found. CH disc samples and MD synovial fluid samples presented, as common proteins, HPX (protection against oxidative stress) and SERPINC1 (biosynthetic pathway of collagen). 

LUM is associated with the regulation of collagen fibers and with cell migration. In this study, LUM was present in all disc samples, and it has been pointed out to be elevated when the disc is under stress, as it enhances tissue repair [31]. Ulmner [32] reported that higher levels of LUM in synovial tissue might diminish TMD surgical success. On the other hand, TNC was present in all disc samples and in DDWoR and MD synovial fluid sample, being an important protein in wound healing [33]. 

Temporomandibular joint discs are fibrocartilaginous discs composed mainly by collagen, glycosaminoglycan and proteoglycans [34]. Studies in human adults and fetuses showed the expression of mainly collagen type I and III in TMJ discs, with type I collagen observed in the posterior band of the articular disc and collagen type III on the inferior surface of the articular disc [35,36]. Moreover, collagen type II synthesis was expressed on the external layer of the TMJ disc [37]. In this study, collagen type IV was identified in MD and CH samples (Table 8), and a previous study observed the presence of collagen type IV in the middle part of fetuses’ TMJ disc, indicating the development of blood vessels [38]. The TMJ disc is an avascular tissue, although under stress it may undergo metaplasia, forming a vascularized fibrous tissue. Collagen type VII was present in all samples, and along with collagen type IV, it has chondroprotective effects against inflammation [39]. Collagen type XII and XIV were present in the disc samples of this study, which have never been identified in this region before in humans. A study identified collagen type XII only in bovine disc samples, which helps maintain collagen type I integrity [40]. Nevertheless, collagen type XIV was also observed in all TMJ disc samples, and it plays an essential structural role in the integrity of collagen type I, mechanical properties, organization, and shape of articular cartilage, which has never been described in the TMJ disc before [41]. This is important information to understand the composition’s strength and weakness of the TMJ disc.

## 5. Conclusions

In conclusion, many proteins were identified for the first time in the TMJ disc and synovial fluid of the groups DDWoR, MD and CH, leading to the enlightenment of each pathology’s etiology, modulation and progression. Further studies with a greater sample are necessary to evaluate other proteins that might be present in these pathologies as well.

## Figures and Tables

**Figure 1 diagnostics-11-00601-f001:**
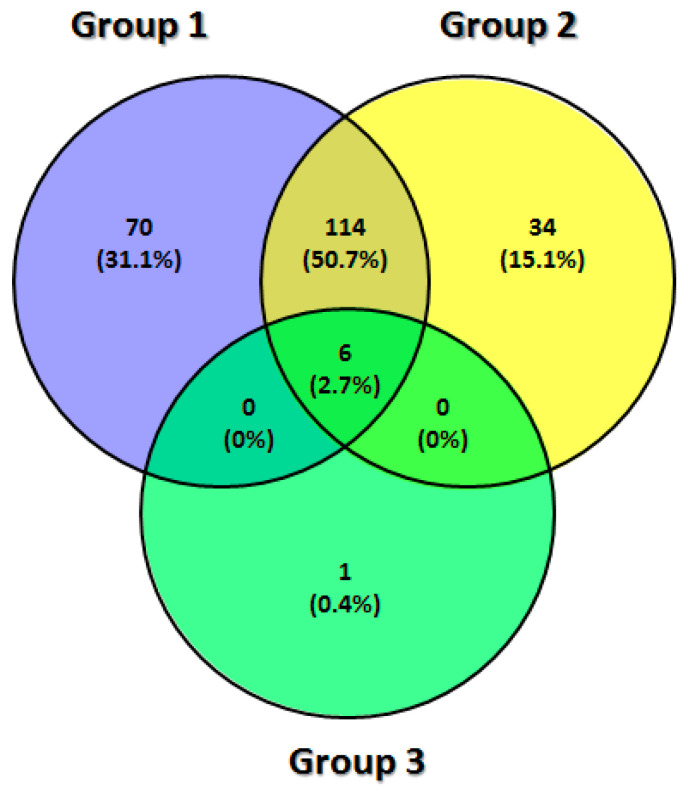
Venn diagram for synovial fluid: group 1—DDWoR, group 2—MD, group 3—CH.

**Figure 2 diagnostics-11-00601-f002:**
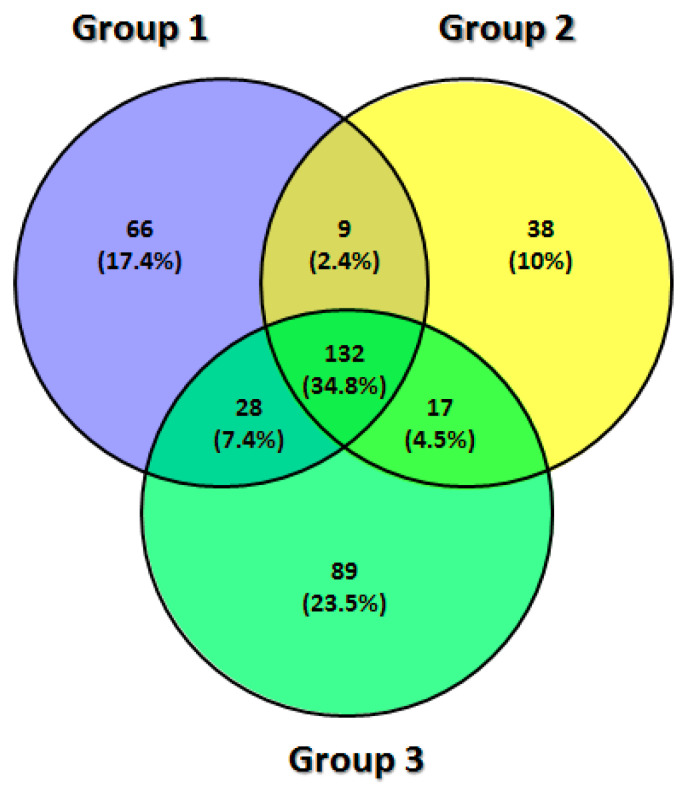
Venn diagram for the TMJ disc: group 1—DDWoR, group 2—MD, group 3—CH.

**Figure 3 diagnostics-11-00601-f003:**
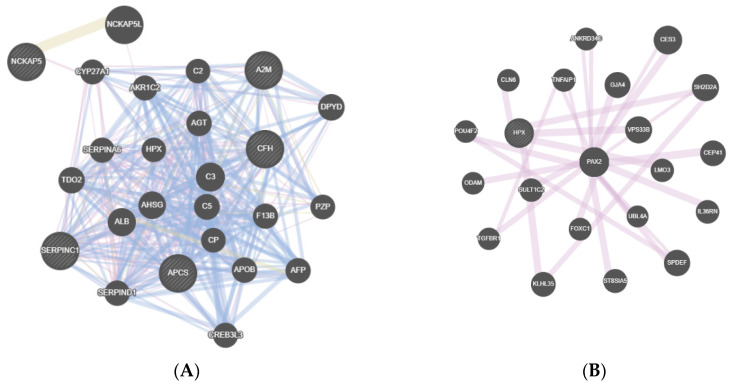
Gene interactions between the main functional proteins of synovial fluid. (**A**) showing the gene interactions of the DDWoR group. (**B**) showing the gene interactions of the MD group. (**C**) showing the gene interactions of the CH group. (**D**) showing the gene interactions of the DDWoR and MD group. (**E**) showing the gene interactions of the DDWoR, MD and CH group.

**Figure 4 diagnostics-11-00601-f004:**
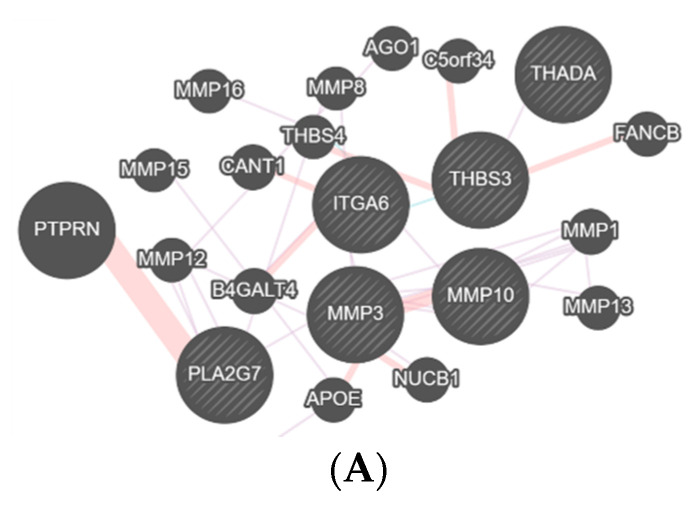
Gene interactions between the main functional proteins of the TMJ disc. (**A**) showing the gene interactions of the DDWoR group. (**B**) showing the gene interactions of the MD group. (**C**) showing the gene interactions of the CH group. (**D**) showing the gene interactions of the CH group. (**E**) showing the gene interactions of the DDWoR and CH group. (**F**) showing the gene interactions of the MD and CH group. (**G**) showing the gene interactions of the DDWoR, MD and CH group.

**Table 1 diagnostics-11-00601-t001:** Baseline characteristics of the sample, showing age and pathology of each female patient.

Number	Age	Diagnostic
1	18	Condylar Hyperplasia
2	20	Condylar Hyperplasia
3	38	Mandibular Dislocation
4	38	Mandibular Dislocation
5	36	Condylar Hyperplasia
6	29	Condylar Hyperplasia
7	25	Disc Displacement Without Reduction
8	25	Disc Displacement Without Reduction
9	52	Disc Displacement Without Reduction

**Table 2 diagnostics-11-00601-t002:** Gene code and name of the proteins expressed in synovial fluid of all groups (disc displacement without reduction (DDWoR), mandibular dislocation (MD), condylar hyperplasia (CH) and between the groups DDWoR and MD, DDWoR and CH, MD and CH and DDWoR, MD and CH.

Protein Expressed in Each Group of TMJ Synovial Fluid Sample (*n* = 225)											
DDWoR (*n* = 70)		MD (*n* = 34)		CH (*n* = 1)		DDWoR and MD (*n* = 114)		DDWoR and CH (*n* = 0)	MD and CH (*n* = 0)	DDWoR, MD and CH (*n* = 6)	
Code	Name	Code	Name	Code	Name	Code	Name	X	X	Code	Name
A2M	Alpha-2-Macroglobulin	ACTR3B	Actin Related Protein 3B	ADH1	Alcohol Dehydrogenase Subunit Alpha	ABI3BP	ABI Family Member 3 Binding Protein			ENO1	Enolase 1
ANXA5	Annexin A5	ACTR3C	Actin Related Protein 3C			ACTA1	Actin Alpha 1, Skeletal Muscle			ENO2	Enolase 2
APCS	Amyloid P Component	AKNA	AT-Hook Transcription Factor			ACTA2	Actin Alpha 2, Smooth Muscle			ENO3	Enolase 3
APOH	Apolipoprotein H	ALDH1L1	Aldehyde Dehydrogenase 1 Family Member L1			ACTB	Actin Beta			MYH16	Myosin Heavy Chain 16 Pseudogene
ARHGAP21	Rho GTPase Activating Protein 21	C4A	Complement C4A (Rodgers Blood Group)			ACTBL2	Actin Beta Like 2			RPL7L1	Ribosomal Protein L7 Like 1
CFH	Complement Factor H	C4B_2	Complement Component 4B			ACTC1	Actin Alpha Cardiac Muscle 1			SHLD3	Shieldin Complex Subunit 3
CHD8	Chromodomain Helicase DNA Binding Protein 8	C7orf57	Complement C7			ACTG1	Actin Gamma 1				
CILP2	Cartilage Intermediate Layer Protein	CAGE1	Cancer Antigen 1			ACTG2	Actin Gamma 2, Smooth Muscle				
CNOT6L	CCR4-NOT Transcription Complex Subunit 6 Like	CPSF2	Cleavage And Polyadenylation Specific Factor 2			ALB	Albumin				
DAGLA	Diacylglycerol Lipase Alpha	DCAF4L2	DDB1 And CUL4 Associated Factor 4 Like 2			ANXA1	Annexin A1				
DPYSL2	Dihydropyrimidinase Like 2	DHRS11	Dehydrogenase/Reductase 11			ANXA2	Annexin A2				
DPYSL3	Dihydropyrimidinase Like 3	DMD	Dystrophin			ANXA2P2	Annexin A2 Pseudogene 2				
DYM	Dymeclin	FLNA	Filamin A			APOA1	Apolipoprotein A1				
DYNC1H1	Dynein Cytoplasmic 1 Heavy Chain	HPR	Haptoglobin-Related Protein			ASPN	Asporin				
ENPP3	Ectonucleotide Pyrophosphatase/Phosphodiesterase 3	HPX	Hemopexin			ATP5F1B	ATP Synthase F1 Subunit Beta				
FGFR2	Fibroblast Growth Factor Receptor 2	IFT122	Intraflagellar Transport 122			BGN	Biglycan				
GPSM2	G Protein Signaling Modulator 2	LMO7	LIM Domain 7			C3	Complement C3				
GPX3	Glutathione Peroxidase 3	MYO6	Myosin VI			CILP	Cartilage Intermediate Layer Protein				
GSTP1	Glutathione S-Transferase Pi 1	PDIA3	Protein Disulfide Isomerase Family A Member 3			CLU	Clusterin				
H2BC1	H2B Clustered Histone 1	PPFIA1	PTPRF Interacting Protein Alpha 1			COL12A1	Collagen Type XII Alpha 1 Chain				
H2BE1	H2B.E Variant Histone 1	PPFIA2	PTPRF Interacting Protein Alpha 2			COL14A1	Collagen Type XIV Alpha 1 Chain				
HSPA1A	Heat Shock Protein Family A (Hsp70) Member 1A	PRDX1	Peroxiredoxin 1			COL1A1	Collagen Type I Alpha 1 Chain				
HSPA1B	Heat Shock Protein Family A (Hsp70) Member 1B	PRDX2	Peroxiredoxin 2			COL6A1	Collagen Type VI Alpha 1 Chain				
HSPA1L	Heat Shock Protein Family A (Hsp70) Member 1 Like	RGMB	Repulsive Guidance Molecule BMP Co-Receptor B			COL6A2	Collagen Type VI Alpha 2 Chain				
HSPA2	Heat Shock Protein Family A (Hsp70) Member 2	SACM1L	SAC1 Like Phosphatidylinositide Phosphatase			COL6A3	Collagen Type VI Alpha 3 Chain				
HSPA8	Heat Shock Protein Family A (Hsp70) Member 8	SERPINA9	Serpin Family A Member 9			COMP	Thrombospondin-5				
IGLC1	Immunoglobulin Lambda Constant 1	SERPINH1	Serpin Family H Member 1			DCN	Decorin				
IGLC2	Immunoglobulin Lambda Constant	SLC4A1	Solute Carrier Family 4 Member 1			DES	Desmin				
IGLC3	Immunoglobulin Lambda Constant 3	SMPD3	Sphingomyelin Phosphodiesterase 3			DPT	Dermatopontin				
	Immunoglobulin Lambda Constant 6		Teneurin Transmembrane Protein 4				Fibrillin 1				
IGLC6	Immunoglobulin Lambda Constant 7	TENM4	Transmembrane O-Mannosyltransferase Targeting Cadherins 3			FBN1	Fibrinogen Alpha Chain				
IGLC7	Immunoglobulin Lambda Like Polypeptide 1	TMTC3	Testis Specific 10			FGA	Fibrinogen Beta Chain				
IGLL1	Immunoglobulin Lambda Like Polypeptide 5	TSGA10	Transthyretin			FGB	Fibrinogen Gamma Chain				
IGLL5	Interferon Regulatory Factor 7	TTR	Ubiquitin Specific Peptidase 10			FGG	Fibromodulin				
IRF7	Kalirin RhoGEF Kinase	USP10	Actin Related Protein 3B			FMOD	Fibronectin 1				
KALRN	Kelch Repeat And BTB Domain Containing 11					FN1	Glyceraldehyde-3-Phosphate Dehydrogenase				
KBTBD11	Keratocan					GAPDH	Gelsolin				
KERA	Keratin 18					GSN	H2B Clustered Histone 11				
KRT18	Keratin 7					H2BC11	H2B Clustered Histone 12				
KRT7	Keratin 8					H2BC12	H2B Clustered Histone 13				
KRT8	Keratin 84					H2BC13	H2B Clustered Histone 14				
KRT84	Putative Uncharacterized Protein					H2BC14	H2B Clustered Histone 15				
LOC400499	Leucine Rich Repeat Containing 9					H2BC15	H2B Clustered Histone 17				
LRRC9	Mitogen-Activated Protein Kinase Kinase Kinase 7					H2BC17	H2B Clustered Histone 18				
MAP3K7	Microfibril Associated Protein 5					H2BC18	H2B Clustered Histone 21				
MFAP5	Myosin Light Chain 6B					H2BC21	H2B Clustered Histone 3				
MYL6B	NCK Associated Protein 5					H2BC3	H2B Clustered Histone 5				
NCKAP5	Nik Related Kinase					H2BC5	H2B Clustered Histone 9				
NRK	Pericentriolar Material 1					H2BC9	H2B.S Histone 1				
PCM1	Procollagen C-Endopeptidase Enhancer					H2BS1	H2B.U Histone 1				
PCOLCE	RAD54 Like					H2BU1	Hemoglobin Subunit Alpha 1				
RAD54L	Retinol Dehydrogenase 5					HBA1	Hemoglobin Subunit Alpha 2				
RDH5	Ret Proto-Oncogene					HBA2	Hemoglobin Subunit Beta				
RET	Regulatory Factor X1					HBB	Hemoglobin Subunit Delta				
RFX1	RPTOR Independent Companion Of MTOR Complex 2					HBD	Hemoglobin Subunit Epsilon 1				
RICTOR	RIMS Binding Protein 3					HBE1	Hemoglobin Subunit Gamma 1				
RIMBP3	RUN And FYVE Domain Containing 2					HBG1	Hemoglobin Subunit Gamma 2				
RUFY2	Serpin Family C Member 1					HBG2	Haptoglobin				
SERPINC1	Serpin Family F Member 1					HP	Heat Shock Protein Family B (Small) Member 1				
SERPINF1	SEC14 And Spectrin Domain Containing 1					HSPB1	Immunoglobulin Heavy Constant Alpha 1				
SESTD1	Small Nuclear Ribonucleoprotein U5 Subunit 200					IGHA1	Immunoglobulin Heavy Constant Alpha 2 (A2m Marker)				
SNRNP200	SVOP Like					IGHA2	Immunoglobulin Heavy Constant Gamma 1 (G1m Marker				
SVOPL	Transcription Elongation Factor, Mitochondrial					IGHG1	Immunoglobulin Heavy Constant Gamma 2				
TEFM	Thrombospondin 3					IGHG2	Immunoglobulin Heavy Constant Gamma 3				
THBS3	Tenascin C					IGHG3	Immunoglobulin Heavy Constant Gamma 4				
TNC	Trio Rho Guanine Nucleotide Exchange Factor					IGHG4	Immunoglobulin Kappa Constant				
TRIO	Tubulin Beta 1 Class VI					IGKC	Internexin Neuronal Intermediate Filament Protein Alpha				
TUBB1	Ubiquitin Specific Peptidase 42					INA	Galectin 1				
USP42	WW Domain Binding Protein 1 Like					LGALS1	Lamin				
WBP1L	Zinc Finger ZZ-Type And EF-Hand Domain Containing 1					LMNA	Lumican				
ZZEF1	H2B Clustered Histone 1					LUM	Microfibril Associated Protein 4				
						MFAP4	Myosin Light Chain 6				
						MYL6	Myocilin				
						MYOC	Neurofilament Heavy				
						NEFH	Neurofilament Light				
						NEFL	Neurofilament Medium				
						NEFM	Osteoglycin				
						OGN	Pellino E3 Ubiquitin Protein Ligase Family Member 3				
						PELI3	Pyruvate Kinase M1/2				
						PKM	POTE Ankyrin Domain Family Member E				
						POTEE	POTE Ankyrin Domain Family Member F				
						POTEF	POTE Ankyrin Domain Family Member I				
						POTEI	POTE Ankyrin Domain Family Member J				
						POTEJ	POTE Ankyrin Domain Family Member K, Pseudogene				
						POTEKP	Peptidylprolyl Isomerase A				
						PPIA	Proline And Arginine Rich End Leucine Rich Repeat Protein				
						PRELP	Peripherin				
						PRPH	S100 Calcium Binding Protein A10				
						S100A10	Serpin Family A Member 1				
						SERPINA1	Superoxide Dismutase 3				
						SOD3	Transferrin				
						TF	Transforming Growth Factor Beta Induced				
						TGFBI	Thrombospondin 4				
						THBS4	Tenascin XA				
						TNXA	Tenascin XB				
						TNXB	Tubulin Alpha 1a				
						TUBA1A	Tubulin Alpha 1b				
						TUBA1B	Tubulin Alpha 1c				
						TUBA1C	Tubulin Alpha 3c				
						TUBA3C	Tubulin Alpha 3d				
						TUBA3D	Tubulin Alpha 3e				
						TUBA3E	Tubulin Alpha 4a				
						TUBA4A	Tubulin Alpha 8				
						TUBA8	Tubulin Beta Class I				
						TUBB	Tubulin Beta 2A Class IIa				
						TUBB2A	Tubulin Beta 2B Class IIb				
						TUBB2B	Tubulin Beta 3 Class III				
						TUBB3	Tubulin Beta 4A Class IVa				
						TUBB4A	Tubulin Beta 4B Class IVb				
						TUBB4B	Tubulin Beta 6 Class V				
						TUBB6	Tubulin Beta 8 Class VIII				
						TUBB8	Tubulin Beta 8B				
						TUBB8B	Versican				
						VCAN	VIM				
						VIM	ABI Family Member 3 Binding Protein				

**Table 3 diagnostics-11-00601-t003:** Gene code and name of the proteins expressed in temporomandibular joint (TMJ) discs of all groups (DDWoR, MD, CH) and between the groups DDWoR and MD, DDWoR and CH, MD and CH and DDWoR, MD and CH.

Protein Expressed in Each Group of TMJ Disc Sample (*n* = 379)													
DDWoR (*n*= 66)		MD (*n* = 38)		CH (*n* = 89)		DDWoR and MD (*n* = 9)		DDWoR and CH (*n* = 28)		MD and CH (*n* = 17)		DDWoR, MD and CH (*n* = 132)	
Code	Name	Code	Name	Code	Name	Code	Name	Code	Name	Code	Name	Code	Name
ABCC9	ATP Binding Cassette Subfamily C Member 9	AFTPH	Aftiphilin	ACTN1	Actinin Alpha 1	ATP7B	ATPase Copper Transporting Beta	ACAN	Aggrecan	ATP5F1B	ATP Synthase F1 Subunit Beta	A2M	Alpha-2-Macroglobulin
ACSS3	Acyl-CoA Synthetase Short Chain Family Member 3	AKAP13	A-kinase anchor protein 13	ACTN4	Actinin Alpha 4	AXIN2	Axin 2	APOH	Apolipoprotein H	GFAP	Glial Fibrillary Acidic Protein	ABI3BP	ABI Family Member 3 Binding Protein
AGO4	Argonaute RISC Component 4	ALDH3A2	Aldehyde dehydrogenase family 3 member A2	ACTR3	Actin Related Protein 3	C4A	Complement C4A	BRD3	Bromodomain Containing 3	KRT3	Keratin 3	ACTA1	Actin Alpha 1, Skeletal Muscle
AMBP	Alpha-1-Microglobulin/Bikunin Precursor	ANKRD44	Serine/threonine-protein phosphatase 6 regulatory ankyrin repeat subunit B	ADAM10	ADAM Metallopeptidase Domain 10	C4B	Complement C4B	CLTC	Clathrin Heavy Chain	KRT5	Keratin 5	ACTA2	Actin Alpha 2, Smooth Muscle
ANKRD17	Ankyrin Repeat Domain 17	ANKRD52	Serine/threonine-protein phosphatase 6 regulatory ankyrin repeat subunit C	ADSL	Adenylosuccinate Lyase	C4B_2	Complement Component 4B	COL1A1	Collagen Type I Alpha 1 Chain	KRT6A	Keratin 6A	ACTB	Actin Beta
ARHGAP35	Rho GTPase Activating Protein 35	ARMH3	Armadillo-like helical domain-containing protein 3	ALDOA	Aldolase, Fructose-Bisphosphate A	KERA	Keratocan	COL4A6	Collagen Type IV Alpha 6 Chain	KRT6B	Keratin 6B	ACTBL2	Actin Beta Like 2
ARHGEF10	Rho Guanine Nucleotide Exchange Factor 10	CCDC88A	Girdin	ALDOC	Aldolase, Fructose-Bisphosphate C	KIAA0556	Katanin Interacting Protein	DNAH8	Defensin Alpha 1	KRT6C	Keratin 6C	ACTC1	Actin Alpha Cardiac Muscle 1
ATAD2B	ATPase Family AAA Domain Containing 2B	CLUH	Clustered mitochondria protein homolog	ANKMY1	Ankyrin Repeat And MYND Domain Containing 1	MAP4	Microtubule Associated Protein 4	EEF1A1	Dynein Axonemal Heavy Chain 8	KRT75	Keratin 75	ACTG1	Actin Gamma 1
BCAS2	BCAS2 Pre-MRNA Processing Factor	COL4A1	Collagen alpha-1(IV) chain	ANXA5	Annexin A5	SEMA4F	Semaphorin 4F	EEF1A1P5	Eukaryotic Translation Elongation Factor 1 Alpha 1	KRT76	Keratin 76	ACTG2	Actin Gamma 2
CARNS1	Carnosine Synthase 1	DOCK10	Dedicator of cytokinesis protein 10	ANXA6	Annexin A6			EEF1A2	Eukaryotic Translation Elongation Factor 1 Alpha 1 Pseudogene 5	KRT78	Keratin 78	ALB	Albumin
CCDC187	Coiled-Coil Domain Containing 187	DTHD1	Death domain-containing protein 1	ASXL1	ASXL Transcriptional Regulator 1			HMCN2	Eukaryotic Translation Elongation Factor 1 Alpha 2	KRT79	Keratin 79	ANXA1	Annexin A1
CDCP1	CUB Domain Containing Protein 1	ERAS	GTPase ERas	ATP2C1	ATPase Secretory Pathway Ca2+ Transporting 1			HSPA2	Hemicentin 2	KRT81	Keratin 81	ANXA2	Annexin A2
CDH3	Cadherin 3	ERBIN	Erbin	BLOC1S1	Biogenesis Of Lysosomal Organelles Complex 1 Subunit 1			HSPA8	Heat Shock Protein Family A (Hsp70) Member 2	KRT83	Keratin 83	ANXA2P2	Annexin A2 Pseudogene 2
CHD7	Chromodomain Helicase DNA Binding Protein 7	FLNA	Filamin-A	BRCA2	BRCA2 DNA Repair Associated			HYDIN	Heat Shock Protein Family A (Hsp70) Member 8	KRT85	Keratin 85	APCS	Amyloid P Component
CHD8	Chromodomain Helicase DNA Binding Protein 8	GOT1L1	Putative aspartate aminotransferase, cytoplasmic 2	CABP5	Calcium Binding Protein 5			IGLC1	HYDIN Axonemal Central Pair Apparatus Protein	KRT86	Keratin 86	APOA1	Apolipoprotein A1
CHD9	Chromodomain Helicase DNA Binding Protein 9	HHLA1	HERV-H LTR-associating protein 1	CACNA2D3	Calcium Voltage-Gated Channel Auxiliary Subunit Alpha2delta 3			IGLC2	Immunoglobulin Lambda Constant 1	PKM	Pyruvate Kinase M1/2	ASPN	Asporin
CSTF2T	Cleavage Stimulation Factor Subunit 2 Tau Variant	IGHV3OR16–9	Immunoglobulin heavy variable 3/OR16–9 (non-functional)	CCDC18	Coiled-Coil Domain Containing 18			IGLC3	Immunoglobulin Lambda Constant 2	TTBK2	Tau Tubulin Kinase 2	BGN	Biglycan
ECH1	Enoyl-CoA Hydratase 1	KDF1	Keratinocyte differentiation factor 1	CDC20	Cell Division Cycle 20			IGLC6	Immunoglobulin Lambda Constant 3			C3	Complement C3
ELAVL3	ELAV Like RNA Binding Protein 3	L1CAM	Neural cell adhesion molecule L1	CENPF	Centromere Protein F			IGLC7	Immunoglobulin Lambda Constant 6			CILP	Cartilage Intermediate Layer Protein
EML4	EMAP Like 4	MARK1	Serine/threonine-protein kinase MARK1	CFAP20DC	CFAP20 Domain Containing			IGLL1	Immunoglobulin Lambda Constant 7			CILP2	Cartilage Intermediate Layer Protein 2
FARP2	FERM, ARH/RhoGEF And Pleckstrin Domain Protein 2	NEIL3	Endonuclease 8-like 3	CNTN1	Contactin 1			IGLL5	Immunoglobulin Lambda Like Polypeptide 1			CLU	Clusterin
FBN1	Fibrillin 1	NOL8	Nucleolar protein 8	COQ8B	Coenzyme Q8B			LOC441081	Immunoglobulin Lambda Like Polypeptide 5			COL12A1	Collagen Type XII Alpha 1 Chain
GALK2	Galactokinase 2	NUFIP1	Nuclear fragile X mental retardation-interacting protein 1	CTNNA3	Catenin Alpha 3			MIS18BP1	POM121 Membrane Glycoprotein (Rat) Pseudogene			COL14A1	Collagen Type XIV Alpha 1 Chain
GPR162	G Protein-Coupled Receptor 162	NUMA1	Nuclear mitotic apparatus protein 1	DPYSL2	Dihydropyrimidinase Like 2			MYO15B	MIS18 Binding Protein 1			COL6A1	Collagen Type VI Alpha 1 Chain
GPRASP1	G Protein-Coupled Receptor Associated Sorting Protein 1	PARP10	Protein mono-ADP-ribosyltransferase PARP10	EHD2	EH Domain Containing 2			POSTN	Myosin XVB			COL6A2	Collagen Type VI Alpha 2 Chain
IKBKE	Inhibitor Of Nuclear Factor Kappa B Kinase Subunit Epsilon	PCDHA4	Protocadherin alpha-4	EYS	Eyes Shut Homolog			SERPINA9	Periostin			COL6A3	Collagen Type VI Alpha 3 Chain
INS	Insulin	POLD1	DNA polymerase delta catalytic subunit	F13A1	Coagulation Factor XIII A Chain			VTN	Serpin Family A Member 9			COMP	Cartilage Oligomeric Matrix Protein
IRF2BPL	Interferon Regulatory Factor 2 Binding Protein Like	POM121L2	POM121-like protein 2	GOLGA4	Golgin A4							DCN	Decorin
ITGA6	Integrin Subunit Alpha 6	PPFIA1	Liprin-alpha-1	GSTP1	Glutathione S-Transferase Pi 1							DES	Desmin
KRT26	Keratin 26	PPFIA2	Liprin-alpha-2	GVINP1	GTPase, Very Large Interferon Inducible Pseudogene 1							DMD	Dystrophin
LEMD2	LEM Domain Nuclear Envelope Protein 2	PRR14L	Protein PRR14L	H3-2	H3.2 Histone (Putative)							DPT	Dermatopontin
MAP3K21	Mitogen-Activated Protein Kinase Kinase Kinase 21	PTPN7	Tyrosine-protein phosphatase non-receptor type 7	H3-3A	H3.3 Histone A							ENO1	Enolase 1
MDGA1	MAM Domain Containing Glycosylphosphatidylinositol Anchor 1	RASSF10	Ras association domain-containing protein 10	H3-3B	H3.3 Histone B							ENO2	Enolase 2
MMP10	Matrix Metallopeptidase 10	RPS6KA6	Ribosomal protein S6 kinase alpha-6	H3-4	H3.4 Histone							ENO3	Enolase 3
MMP27	Matrix Metallopeptidase 27	TRIO	TRIO and F-actin-binding protein	H3-5	H3.5 Histone							FBLN1	Fibulin 1
MMP3	Matrix Metallopeptidase 3	TSC1	Hamartin	HEATR6	HEAT Repeat Containing 6							FGA	Fibrinogen Alpha Chain
MOS	MOS Proto-Oncogene, Serine/Threonine Kinas	UPK3A	Uroplakin-3a	HPX	Hemopexin							FGB	Fibrinogen Beta Chain
MYL6	Myosin Light Chain 6	UROD	Uroporphyrinogen decarboxylase	HSP90B1	Heat Shock Protein 90 Beta Family Member 1							FGG	Fibrinogen Gamma Chain
MYO7B	Myosin VIIB			HSPA1A	Heat Shock Protein Family A (Hsp70) Member 1A							FLNB	Filamin B
NT5E	5’-Nucleotidase Ecto			HSPA1B	Heat Shock Protein Family A (Hsp70) Member 1B							FMOD	Fibromodulin
OLFML1	Olfactomedin Like 1			HSPA1L	Heat Shock Protein Family A (Hsp70) Member 1 Like							FN1	Fibronectin 1
PGM5	Phosphoglucomutase 5			HSPA5	Heat Shock Protein Family A (Hsp70) Member 5							GAPDH	Glyceraldehyde-3-Phosphate Dehydrogenase
PHKA2	Phosphorylase Kinase Regulatory Subunit Alpha 2			IGFN1	Immunoglobulin Like And Fibronectin Type III Domain Containing 1							GPX3	Glutathione Peroxidase 3
PLA2G7	Phospholipase A2 Group VII			INF2	Inverted Formin 2							GSN	Angiotensin I Converting Enzyme 2
POR	Cytochrome P450 Oxidoreductase			L3MBTL4	L3MBTL Histone Methyl-Lysine Binding Protein 4							H2BC1	H2B Clustered Histone 1
RANBP17	RAN Binding Protein 17			LMNB1	Lamin B1							H2BC11	H2B Clustered Histone 11
RGS22	Regulator Of G Protein Signaling 22			LMNB2	Lamin B2							H2BC12	H2B Clustered Histone 12
RIF1	Replication Timing Regulatory Factor 1			MFAP5	Microfibril Associated Protein 5							H2BC13	H2B Clustered Histone 13
RTN4	Reticulon 4			MRPL50	Mitochondrial Ribosomal Protein L50							H2BC14	H2B Clustered Histone 14
SARS2	Seryl-TRNA Synthetase 2, Mitochondrial			MS4A6A	Membrane Spanning 4-Domains A6A							H2BC15	H2B Clustered Histone 15
SEPHS2	Selenophosphate Synthetase 2			MUC4	Mucin 4, Cell Surface Associated							H2BC17	H2B Clustered Histone 17
SLFN13	Schlafen Family Member 13			MYH14	Myosin Heavy Chain 14							H2BC18	H2B Clustered Histone 18
SLK	STE20 Like Kinase			MYL6B	Myosin Light Chain 6B							H2BC21	H2B Clustered Histone 21
SPATA20	Spermatogenesis Associated 20			NEK10	NIMA Related Kinase 10							H2BC3	H2B Clustered Histone 3
SPATA5	Spermatogenesis Associated 5			PAK3	P21 (RAC1) Activated Kinase 3							H2BC5	H2B Clustered Histone 5
SPTA1	Spectrin Alpha, Erythrocytic 1			PAPOLA	Poly(A) Polymerase Alpha							H2BC9	H2B Clustered Histone 9
SQLE	Squalene Epoxidase			PAPOLG	Poly(A) Polymerase Gamma							H2BS1	H2B.S Histone 1
ST20-AS1	ST20 Antisense RNA 1			PDIA3	Protein Disulfide Isomerase Family A Member 3							H2BU1	H2B.U Histone 1
STIL	STIL Centriolar Assembly Protein			PDLIM4	PDZ And LIM Domain 4							HBA1	Hemoglobin Subunit Alpha 1
TACC2	Transforming Acidic Coiled-Coil Containing Protein 2			RALBP1	RalA Binding Protein 1							HBA2	Hemoglobin Subunit Alpha 2
TAP1	Transporter 1, ATP Binding Cassette Subfamily B Member			RNF213	Ring Finger Protein 213							HBB	Hemoglobin Subunit Beta
THADA	THADA Armadillo Repeat Containing			SBF2	SET Binding Factor 2							HBD	Hemoglobin Subunit Delta
THBS3	Thrombospondin 3			SERPINF1	Serpin Family F Member 1							HBE1	Hemoglobin Subunit Epsilon 1
UQCRC1	Ubiquinol-Cytochrome C Reductase Core Protein 1			SERPINH1	Serpin Family H Member 1							HBG1	Hemoglobin Subunit Gamma 1
VWA3A	Von Willebrand Factor A Domain Containing 3A			SLC4A5	Solute Carrier Family 4 Member 5							HBG2	Hemoglobin Subunit Gamma 2
ZNF333	Zinc Finger Protein 333			SLIT2	Slit Guidance Ligand 2							HBZ	Hemoglobin Subunit Zeta
				SMPD3	Sphingomyelin Phosphodiesterase 3							HP	Haptoglobin
				TAPT1	Transmembrane Anterior Posterior Transformation 1							HPR	Haptoglobin-Related Protein
				TBX22	T-Box Transcription Factor 22							HSPB1	Heat Shock Protein Family B (Small) Member 1
				TDRD1	Tudor Domain Containing 1							IGHA1	Immunoglobulin Heavy Constant Alpha 1
				TENM4	Teneurin Transmembrane Protein 4							IGHA2	Immunoglobulin Heavy Constant Alpha 2 (A2m Marker)
				THBS1	Thrombospondin 1							IGHG1	Immunoglobulin Heavy Constant Gamma 1 (G1m Marker)
				TJP2	Tight Junction Protein 2							IGHG2	Immunoglobulin Heavy Constant Gamma 2 (G2m Marker)
				TTR	Transthyretin							IGHG3	Immunoglobulin Heavy Constant Gamma 3 (G3m Marker)
				UBP1	Upstream Binding Protein 1							IGHG4	Immunoglobulin Heavy Constant Gamma 4 (G4m Marker)
				WHRN	Whirlin							IGKC	Immunoglobulin Kappa Constant
				ZNF155	Zinc Finger Protein 155							INA	Internexin Neuronal Intermediate Filament Protein Alpha
				ZNF221	Zinc Finger Protein 221							KRT7	Keratin 7
												KRT8	Keratin 8
												KRT84	Keratin 84
												LGALS1	Galectin 1
												LMNA	Lamin A/C
												LUM	Lumican
												MFAP4	Microfibril Associated Protein 4
												MFGE8	Milk Fat Globule EGF And Factor V/VIII Domain Containing
												MYH16	Myosin Heavy Chain 16 Pseudogene
												MYOC	Myocilin
												NEFH	Neurofilament Heavy
												NEFL	Neurofilament Light
												NEFM	Neurofilament Medium
												OGN	Osteoglycin
												POTEE	POTE Ankyrin Domain Family Member E
												POTEF	POTE Ankyrin Domain Family Member F
												POTEI	POTE Ankyrin Domain Family Member I
												POTEJ	POTE Ankyrin Domain Family Member J
												POTEKP	POTE Ankyrin Domain Family Member K, Pseudogene
												PPIA	Peptidylprolyl Isomerase A
												PRDX1	Peroxiredoxin 1
												PRDX2	Peroxiredoxin 2
												PRELP	Proline And Arginine Rich End Leucine Rich Repeat Protein
												PRPH	Peripherin
												RPL7L1	Ribosomal Protein L7 Like 1
												S100A10	S100 Calcium Binding Protein A10
												SALL3	Spalt Like Transcription Factor 3
												SERPINA1	Serpin Family A Member
												SHLD3	Shieldin Complex Subunit 3
												SLC4A1	Solute Carrier Family 4 Member 1
												SOD3	Superoxide Dismutase 3
												TF	Transferrin
												TGFBI	Transforming Growth Factor Beta Induced
												THBS4	Thrombospondin 4
												TNC	Tenascin C
												TNXA	Tenascin XA (Pseudogene)
												TNXB	Tenascin XB
												TUBA1A	Tubulin Alpha 1a
												TUBA1B	Tubulin Alpha 1b
												TUBA1C	Tubulin Alpha 1c
												TUBA3E	Tubulin Alpha 3e
												TUBA4A	Tubulin Alpha 4a
												TUBA8	Tubulin Alpha 8
												TUBB	Tubulin Beta Class I
												TUBB1	Tubulin Beta 1 Class VI
												TUBB2A	Tubulin Beta 2A Class IIa
												TUBB2B	Tubulin Beta 2B Class IIb
												TUBB3	Tubulin Beta 3 Class III
												TUBB4A	Tubulin Beta 4A Class IVa
												TUBB4B	Tubulin Beta 4B Class IVb
												TUBB6	Tubulin Beta 6 Class V
												TUBB8	Tubulin Beta 8 Class VIII
												TUBB8B	Tubulin Beta 8B

**Table 4 diagnostics-11-00601-t004:** Proteins expressed in both synovial fluid and TMJ disc samples of each group.

Protein Expressed in Each Group of TMJ Synovial Fluid and Disc Samples (*n* = 11)						
DDWoR (*n*= 2)	MD (*n* = 3)	CH (*n* = 0)	DDWoR and MD (*n* = 0)	DDWoR and CH (*n* = 0)	MD and CH (*n* = 0)	DDWoR, MD and CH (*n* = 6)
CHD8	FLNA					ENO1
MYL6B	PPFIA1					ENO2
	PPFIA2					ENO3
						MYH16
						RPL7L1
						SHLD3

**Table 5 diagnostics-11-00601-t005:** Gene code, protein name and function for each sample of TMJ synovial fluid.

Synovial Fluid Sample		
Code	Name	Function
**DDWoR**		
A2M	Alpha-2-Macroglobulin	Inhibits inflammatory cytokines.
APCS	Amyloid P Component, Serum	Binds to apoptotic cells at an early stage.
GPSM2	G Protein Signaling Modulator 2	Involved in the development of normal hearing.
KRT18	Keratin 18	Is involved in interleukin-6-mediated barrier protection.
MAP3K7	Mitogen-Activated Protein Kinase Kinase Kinase 7	Mediates signal transduction various cytokines including interleukin-1, transforming growth factor-beta, bone morphogenetic protein 2 and 4, Toll-like receptors, tumor necrosis factor receptor CD40 and B-cell receptor.
SERPINC1	Serpin Family C Member 1	This protein inhibits thrombin and it regulates the blood coagulation cascade.
**MD**		
ALDH1L1	Aldehyde Dehydrogenase 1 Family Member L1	Associated with decreased apoptosis, increased cell motility, and cancer progression.
C4A	Complement C4A (Rodgers Blood Group)	An antimicrobial peptide and a mediator of local inflammation.
HPX	Hemopexin	Acute phase protein that transports heme from the plasma to the liver and may be involved in protecting cells from oxidative stress.
IFT122	Intraflagellar Transport 122	Involved in cell cycle progression, signal transduction, apoptosis, and gene regulation.
MYO6	Myosin VI	This protein maintains the structural integrity of inner ear hair cells and mutations in this gene cause hearing loss.
PRDX1	Peroxiredoxin 1	Has an antioxidant protective role in cells and may contribute to the antiviral activity of CD8(+) T-cells.
SERPINH1	Serpin Family H Member 1	Plays a role in collagen biosynthesis as a collagen-specific molecular chaperone.
SMPD3	Sphingomyelin Phosphodiesterase 3	Mediates cellular functions, such as apoptosis and growth arrest.
**CH**		
ADH1	Alcohol Dehydrogenase Subunit Alpha	Catalyzes the oxidation of alcohols to aldehydes.
DDWoR and MD		
ANXA1	Annexin A1	Inhibits phospholipase A2 and has anti-inflammatory activity.
ANXA2	Annexin A2	Functions as an autocrine factor which heightens osteoclast formation and bone resorption.
ASPN	Asporin	Regulate chondrogenesis by inhibiting transforming growth factor-beta 1-induced gene expression in cartilage. May induce collagen mineralization.
BGN	Biglycan	Plays a role in bone growth, muscle development and regeneration, and collagen fibril assembly in multiple tissues. This protein may also regulate inflammation and innate immunity.
CILP	Cartilage Intermediate Layer Protein	This protein is present in the cartilage intermediate layer protein (CILP), which increases in early osteoarthrosis cartilage.
CLU	Clusterin	Under stress conditions can be found in the cell cytosol. May be involved in cell death, tumor progression, and neurodegenerative disorders
COMP	Thrombospondin-5	Present in rheumatoid arthritis, is a noncollagenous extracellular matrix protein.
DCN	Decorin	Has a stimulatory effect on autophagy and inflammation and an inhibitory effect on angiogenesis and tumorigenesis.
FMOD	Fibromodulin	May also regulate TGF-beta activities by sequestering TGF-beta into the extracellular matrix.
FN1	Fibronectin 1	Fibronectin is involved in cell adhesion and migration processes including embryogenesis, wound healing, blood coagulation, host defense.
IGHG1	Immunoglobulin Heavy Constant Gamma 1 (G1m Marker)	Involved in pathways of Interleukin-4 and 13 signaling and IL4-mediated signaling events.
**DDWoR and CH**		
x	x	x
**MD and CH**		
x	x	x
**DDWoR, MD and CH**		
ENO2	Enolase 2	Found in mature neurons and cells of neuronal origin.
ENO3	Enolase 3	May play a role in muscle development and regeneration.

**Table 6 diagnostics-11-00601-t006:** Gene code, protein name and function for each sample of TMJ discs.

Disc Sample		
Code	Name	Function
**DDWoR**		
AMBP	Alpha-1-Microglobulin/Bikunin Precursor	Regulation of the inflammatory process.
MMP10	Matrix Metallopeptidase 10	Breakdown of extracellular matrix.
MMP27	Matrix Metallopeptidase 27	Breakdown of extracellular matrix.
MMP3	Matrix Metallopeptidase 3	Breakdown of extracellular matrix.
PLA2G7	Phospholipase A2 Group VII	Inflammatory and oxidative stress response.
THADA	THADA Armadillo Repeat Containing	Apoptosis pathway.
THBS3	Thrombospondin 3	Matrix interactions.
**MD**		
AKAP13	A-kinase anchor protein 13	Regulation of apoptotic process.
CCDC88A	Girdin	Vascular endothelial growth factor receptor 2 binding.
COL4A1	Collagen alpha-1(IV) chain	Extracellular matrix structural constituent.
ERAS	GTPase ERas	Tumor-like growth properties of embryonic stem cells.
ERBIN	Erbin	Inhibits NOD2-dependent NF-kappa-B signaling and proinflammatory cytokine secretion.
PARP10	Protein mono-ADP-ribosyltransferase PARP10	Negative regulation of fibroblast proliferation.
PPFIA1	Liprin-alpha-1	Cell–matrix adhesion.
PPFIA2	Liprin-alpha-2	Cell–matrix adhesion.
PTPN7	Tyrosine-protein phosphatase non-receptor type 7	Regulation of T and B-lymphocyte development and signal transduction.
UPK3A	Uroplakin-3a	Epithelial cell differentiation.
**CH**		
ACTN4	Actinin Alpha 4	Transcriptional coactivator.
ADAM10	ADAM Metallopeptidase Domain 10	Responsible for the FasL ectodomain shedding.
COQ8B	Coenzyme Q8B	Biosynthesis of coenzyme Q.
HPX	Hemopexin	Protect cells from oxidative stress.
HSPA1A	Heat Shock Protein Family A (Hsp70) Member 1A	Protection of the proteome from stress.
NEK10	NIMA Related Kinase 10	Cellular response to UV irradiation.
PDLIM4	PDZ And LIM Domain 4	Involved in bone development.
SERPINH1	Serpin Family H Member 1	Chaperone in the biosynthetic pathway of collagen.
TTR	Transthyretin	Thyroid hormone-binding protein.
COL1A2	Collagen Type I Alpha 2 Chain	Fibril-forming collagen abundant in bone.
PRG4	Proteoglycan 4	This protein contains both chondroitin sulfate and keratan sulfate glycosaminoglycans.
PTPN13	Protein Tyrosine Phosphatase Non-Receptor Type 13	Regulates negatively FasL induced apoptosis.
**DDWoR and MD**		
C4A	Complement C4A	Antimicrobial peptide and a mediator of local inflammation.
C4B	Complement C4B	Mediator of local inflammation.
C4B_2	Complement Component 4B	Mediator of local inflammatory process.
SEMA4F	Semaphorin 4F	Plays a role in neural development.
Code	Name	Function
**DDWoR and CH**		
ACAN	Aggrecan	Part of the extracellular matrix that withstands compression in cartilage.
COL1A1	Collagen Type I Alpha 1 Chain	Collagen component.
COL4A6	Collagen Type IV Alpha 6 Chain	Major structural component of basement membranes.
HSPA2	Heat Shock Protein Family A (Hsp70) Member 2	Protection of the proteome from stress.
POSTN	Periostin	Extracellular matrix protein that functions in tissue development and regeneration, including wound healing.
**MD and CH**		
KRT6A	Keratin 6A	Epidermis-specific type I keratin involved in wound healing.
DDWoR, MD and CH		
ANXA1	Annexin A1	Anti-inflammatory activity.
ANXA2	Annexin A2	Heightens osteoclast formation and bone resorption.
ANXA2P2	Annexin A2 Pseudogene 2	May be involved in heat-stress response.
APCS	Amyloid P Component	Is involved in dealing with apoptotic cells in vivo.
ASPN	Asporin	Regulates chondrogenesis by inhibiting transforming growth factor-beta 1-induced gene expression in cartilage
BGN	Biglycan	Plays a role in bone growth, and collagen fibril assembly in multiple tissues. This protein may also regulate inflammation and innate immunity.
C3	Complement C3	Modulates inflammation and possesses antimicrobial activity.
CILP	Cartilage Intermediate Layer Protein	Increases in early osteoarthrosis cartilage.
COL12A1	Collagen Type XII Alpha 1 Chain	Type XII collagen.
COL14A1	Collagen Type XIV Alpha 1 Chain	Type XIV collagen.
COL6A1	Collagen Type VI Alpha 1 Chain	Collagen VI.
COL6A2	Collagen Type VI Alpha 2 Chain	Type VI collagen.
COL6A3	Collagen Type VI Alpha 3 Chain	Ttype VI collagen.
COMP	Cartilage Oligomeric Matrix Protein	Degradation of the extracellular matrix.
ENO1	Enolase 1	Tumor suppressor.
ENO2	Enolase 2	Found in mature neurons and cells of neuronal origin.
ENO3	Enolase 3	Plays a role in muscle development and regeneration.
FN1	Fibronectin 1	Involved in wound healing, blood coagulation, host defense.
KRT7	Keratin 7	Co-expressed during differentiation of simple and stratified epithelial tissues.
LUM	Lumican	May regulate collagen fibril organization, epithelial cell migration and tissue repair.
MFAP4	Microfibril Associated Protein 4	Extracellular matrix protein which is involved in cell adhesion or intercellular interactions.
MFGE8	Milk Fat Globule EGF And Factor V/VIII Domain Containing	Promotes phagocytosis of apoptotic cells. This protein has also been implicated in wound healing, autoimmune disease, and cancer.
OGN	Osteoglycin	Induces ectopic bone formation in conjunction with transforming growth factor beta and may regulate osteoblast differentiation.
SOD3	Superoxide Dismutase 3	Antioxidant enzymes that protect tissues from oxidative stress.
TGFBI	Transforming Growth Factor Beta Induced	May be involved in endochondrial bone formation in cartilage.
TNC	Tenascin C	Modulation of inflammatory cytokine.
TNXB	Tenascin XB	Accelerates collagen fibril formation.
VCAN	Versican	A large chondroitin sulfate proteoglycan and is a major component of the extracellular matrix.
VIM	Vimentin	Involved in the stabilization of type I collagen mRNAs for CO1A1 and CO1A2.

**Table 7 diagnostics-11-00601-t007:** Name and function of expressed proteins in common between synovial fluid and TMJ disc sample, and the groups in each protein was expressed.

Name	Function	Disc	Synovial Fluid
Amyloid P Component, Serum	Is involved in dealing with apoptotic cells in vivo.	DDWoR, MD and CH	DDWoR
Annexin A1	Anti-inflammatory activity.	DDWoR, MD and CH	DDWoR and MD
Annexin A2	Heightens osteoclast formation and bone resorption.	DDWoR, MD and CH	DDWoR and MD
Asporin	Regulates chondrogenesis.	DDWoR, MD and CH	DDWoR and MD
Biglycan	Plays a role in bone growth, and collagen fibril assembly in multiple tissues.	DDWoR, MD and CH	DDWoR and MD
Cartilage Intermediate Layer Protein	Increases in early osteoarthrosis cartilage.	DDWoR, MD and CH	DDWoR and MD
Complement C4A	Antimicrobial peptide and a mediator of local inflammation.	DDWoR and MD	MD
Enolase 2	Found in mature neurons and cells of neuronal origin.	DDWoR, MD and CH	DDWoR, MD and CH
Enolase 3	Play a role in muscle development and regeneration.	DDWoR, MD and CH	DDWoR, MD and CH
Fibronectin 1	Involved in wound healing, blood coagulation, host defense.	DDWoR, MD and CH	DDWoR and MD
Hemopexin	Protect cells from oxidative stress.	CH	MD
Lumican	May regulate collagen fibril organization, epithelial cell migration and tissue repair.	DDWoR, MD and CH	DDWoR and MD
Osteoglycin	Regulate osteoblast differentiation.	DDWoR, MD and CH	DDWoR and MD
Serpin Family H Member 1	Chaperones in the biosynthetic pathway of collagen.	CH	MD
Superoxide Dismutase 3	Antioxidant enzymes that protect tissues from oxidative stress.	DDWoR, MD and CH	DDWoR and MD
Tenascin XB	Modulation of inflammatory cytokine.	DDWoR, MD and CH	DDWoR and MD
Transforming Growth Factor Beta Induced	May be involved in endochondral bone formation in cartilage.	DDWoR, MD and CH	DDWoR and MD
Versican	A large chondroitin sulfate proteoglycan and is a major component of the extracellular matrix.	DDWoR, MD and CH	DDWoR and MD

**Table 8 diagnostics-11-00601-t008:** Types of collagen identified in each TMJ disc group.

Type of Collagen Identified in Each Group										
DDWoR	MD		CH		DDWoR and MD	DDWoR and CH		MD and CH	DDWoR, MD and CH	
x	Code	Name	Code	Name	x	Code	Name	x	Code	Name
	COL4A1	Collagen Type IV Alpha 1 Chain	COL1A2	Collagen Type I Alpha 2 Chain		COL1A1	Collagen Type I Alpha 1 Chain		COL12A1	Collagen Type XII Alpha 1 Chain
						COL4A6	Collagen Type IV Alpha 6 Chain		COL14A1	Collagen Type XIV Alpha 1 Chain
									COL6A1	Collagen Type VI Alpha 1 Chain
									COL6A2	Collagen Type VI Alpha 2 Chain
									COL6A3	Collagen Type VI Alpha 3 Chain

## Data Availability

Data is contained within the article.

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
