# Peer review of "Proteomic Expression Profile in Human Temporomandibular Joint Dysfunction"

_diagnostics, 2021, doi:10.3390/diagnostics11040601_

Round 1

Reviewer 1 Report

The authors aimed to evaluate the protein expression profiles in three TMD conditions. Data showed that were also identified different and novel proteins in both synovial fluid and disc sample.

The study is easy to follow and covers an interesting topic, but some minor  issues should be improved before publication. The manuscript needs moderate English change and grammar correction. Please also check typos thorough the text.

Discussion section: In the sentence "Nevertheless, collagen type XIV was also observed in all TMJ disc samples, and it plays an essential structural role in the integrity of collagen type I, mechanical properties, organization, and shape of articular cartilage, which has never been described in TMJ disc before", please add at the end, the appropriate references (please see PubMed ID: 19180004 and PubMed ID: 17452956).

Conclusion Section: This paragraph required a general revision to eliminate redundant sentences and to add some "take-home message".

Author Response

REBUTTAL LETTER

Dear Reviewer,

thank you very much for your considerations. Below there are the changes made in the paper according to the reviewer’s suggestion.

REVIEWER 1 CONSIDERATIONS

The authors aimed to evaluate the protein expression profiles in three TMD conditions. Data showed that were also identified different and novel proteins in both synovial fluid and disc sample.

The study is easy to follow and covers an interesting topic, but some minor  issues should be improved before publication. The manuscript needs moderate English change and grammar correction. Please also check typos thorough the text.

Discussion section: In the sentence "Nevertheless, collagen type XIV was also observed in all TMJ disc samples, and it plays an essential structural role in the integrity of collagen type I, mechanical properties, organization, and shape of articular cartilage, which has never been described in TMJ disc before", please add at the end, the appropriate references (please see PubMed ID: 19180004 and PubMed ID: 17452956).

Conclusion Section: This paragraph required a general revision to eliminate redundant sentences and to add some "take-home message".

ANSWER TO REVIEWER 1:

The English grammar and typos were reviewed as suggested. In line 419 the recommended reference was inserted at the end of the sentence in the discussion section. Conclusion was rewritten to avoid redundant sentences in lines 424-427.

Thank you for your time and the great contributions and suggestions to this paper.

Best regards

Reviewer 2 Report

Dear author,

 the article is well written but i invite you to increase the introduction by inserting the various etiologies of TMD such as inflammatory diseases, trauma, fractures, etc. also add a brief overview of clinical and imaging methods for diagnosis and most popular therapies;

these recent articles may be useful to you:

1) A follow-up study of condyle fracture in children. PMID: 15921889

2) Unilateral fracture of the neck of the upper condyle with dislocation in a child treated with an acrylic splint in the upper arch for functional repositioning of the mandible. PMID: 27398739

3) The efficacy of anterior repositioning splint therapy studied by magnetic resonance imaging. PMID: 12198864

4) Temporomandibular disc displacement with reduction treated with anterior repositioning splint: a 2-year clinical and magnetic resonance imaging (MRI) follow-up. PMID: 32064850

5) Botulinum toxin in the closed treatment of mandibular condylar fracture. PMID: 17452828

6) Occlusal splint therapy and magnetic resonance imaging. PMID: 15615131

7) Polyphenols as potential agents in the management of temporomandibular disorders https://doi.org/10.3390/app10155305

in particular, these papers can be useful for arguing the lines 45-46 regarding disc dislocation (describe the various stages and various therapies) and line 55 when talking about mandibular asymmetries (you could describe the various aetiologies;  conservative and surgical therapies from traditional to more recent)

good luck with your manuscript

Best regards

Author Response

REBUTTAL LETTER

Dear Reviewer,

thank you very much for your considerations. Below there are the changes made in the paper according to the reviewer’s suggestion.

REVIEWER 2 COMMENTS

Dear author,

 the article is well written but i invite you to increase the introduction by inserting the various etiologies of TMD such as inflammatory diseases, trauma, fractures, etc. also add a brief overview of clinical and imaging methods for diagnosis and most popular therapies;

these recent articles may be useful to you:

1) A follow-up study of condyle fracture in children. PMID: 15921889

2) Unilateral fracture of the neck of the upper condyle with dislocation in a child treated with an acrylic splint in the upper arch for functional repositioning of the mandible. PMID: 27398739

3) The efficacy of anterior repositioning splint therapy studied by magnetic resonance imaging. PMID: 12198864

4) Temporomandibular disc displacement with reduction treated with anterior repositioning splint: a 2-year clinical and magnetic resonance imaging (MRI) follow-up. PMID: 32064850

5) Botulinum toxin in the closed treatment of mandibular condylar fracture. PMID: 17452828

6) Occlusal splint therapy and magnetic resonance imaging. PMID: 15615131

7) Polyphenols as potential agents in the management of temporomandibular disorders https://doi.org/10.3390/app10155305

in particular, these papers can be useful for arguing the lines 45-46 regarding disc dislocation (describe the various stages and various therapies) and line 55 when talking about mandibular asymmetries (you could describe the various aetiologies;  conservative and surgical therapies from traditional to more recent)

good luck with your manuscript

ANSWER TO REVIEWER 2

Dear Reviewer,

 thank you very much for your considerations. The English grammar was reviewed as suggested. Also the suggested references were inserted in the introduction section, according to the TMD pathology explanation. In line 35/36 etiology of TMD was described, and new references were inserted. In lines 50-52 treatment options for DDWoR were explained with new references. In line 59-61, the sentence was rewritten to fulfill the suggestions about condyle hyperplasia etiology and treatment. Unfortunately, condylar hyperplasia still has no certain etiology, therefore the aim of this study was to analyze the protein which could contribute to the unveiling its etiology. Also, surgical treatment is mostly indicated to treat condylar hyperplasia, as specified in introduction. Other therapies have not been employed for this condition.

Thank you for your time and the great contributions and suggestions to this paper.

Best regards
